# Hyperbranched Poly(β-amino ester)s (HPAEs) Structure Optimisation for Enhanced Gene Delivery: Non-Ideal Termination Elimination

**DOI:** 10.3390/nano12213892

**Published:** 2022-11-04

**Authors:** Yinghao Li, Zhonglei He, Jing Lyu, Xianqing Wang, Bei Qiu, Irene Lara-Sáez, Jing Zhang, Ming Zeng, Qian Xu, Sigen A, James F. Curtin, Wenxin Wang

**Affiliations:** 1Charles Institute of Dermatology, School of Medicine, University College Dublin, D04 V1W8 Dublin, Ireland; 2BioPlasma Research Group, School of Food Science and Environmental Health, Technological University Dublin, D07 H6K8 Dublin, Ireland; 3State Key Laboratory of Materials-Oriented Chemical Engineering, College of Chemical Engineering, Nanjing Tech University, 30 Puzhu South Road, Nanjing 211816, China; 4Department of Dermatology, The First Affiliated Hospital of Jinan University, Guangzhou Overseas Chinese Hospital, Guangzhou 510630, China; 5Faculty of Engineering and Built Environment, Technological University Dublin, D07 H6K8 Dublin, Ireland

**Keywords:** gene therapy, DNA delivery, transfection, polymeric nano vectors, hyperbranched polymers, poly(β-amino ester)s

## Abstract

Many polymeric gene delivery nano-vectors with hyperbranched structures have been demonstrated to be superior to their linear counterparts. The higher delivery efficacy is commonly attributed to the abundant terminal groups of branched polymers, which play critical roles in cargo entrapment, material-cell interaction, and endosome escape. Hyperbranched poly(β-amino ester)s (HPAEs) have developed as a class of safe and efficient gene delivery vectors. Although numerous research has been conducted to optimise the HPAE structure for gene delivery, the effect of the secondary amine residue on its backbone monomer, which is considered the non-ideal termination, has never been optimised. In this work, the effect of the non-ideal termination was carefully evaluated. Moreover, a series of HPAEs with only ideal terminations were synthesised by adjusting the backbone synthesis strategy to further explore the merits of hyperbranched structures. The HPAE obtained from modified synthesis methods exhibited more than twice the amounts of the ideal terminal groups compared to the conventional ones, determined by NMR. Their transfection performance enhanced significantly, where the optimal HPAE candidates developed in this study outperformed leading commercial benchmarks for DNA delivery, including Lipofectamine 3000, jetPEI, and jetOPTIMUS.

## 1. Introduction

Growing at an impressive pace, gene therapy has emerged as a prospective segment in the biotechnology field and presents a bright outlook, given the increasing investments in R&D and the mounting incidence of genetic disorders [1]. Amid the COVID-19 crisis, the global market for gene therapy, estimated at 1.2 billion dollars in the year 2022, is projected to reach a revised size of 2.7 billion dollars by 2026, according to the Cision PR Newswire. However, the lack of efficient gene delivery vectors still hinders the large-scale manufacturing of gene therapy agents. Viral vectors, such as the adeno-associated virus, lentivirus, retrovirus, and gamma-retrovirus, herpes simplex virus, poxvirus, and vaccinia virus, are the most prevalent gene delivery vectors. Despite their high efficiency, safety concerns, such as severe immune responses and activation of viral components, remain together with a limited payload, compromising their clinical trials [2,3]. Moreover, the high manufacturing cost of viral vectors further impedes their large-scale clinical applications.

Non-viral vectors account for another essential share of the market due to their merits of easy manufacturing and modification according to specific needs. For instance, the world recognised lipid nanoparticles (LNPs) due to their application for the COVID-19 mRNA vaccine vectors. The merits of lipid compositions enable large-scale manufacturing in a short time, but LNP technology still suffers from relatively poor colloidal stability in physical environments and an inflammatory response active potential, leading to safety concerns [4].

On the other hand, polymeric vectors also have the merits of easy operation and modification, indicating a promising candidate for future gene delivery vector development [5]. Among polymeric vectors, poly(β-amino ester)s (PAEs) derivatives are a type of vital gene carrier. Since the first report by Langer et al. in 2000, more than 2500 Linear PAEs have been synthesised and applied in gene therapy [6,7]. In recent decades, numerous works have been conducted to study the polymer structure-transfection performance relationship and optimise the PAE structures. In 2007, Anderson et al. proved that the terminal groups are one of the most critical factors for an efficient vector design [8]. By changing the end-modified groups, the transfection performance can be improved by several magnitude orders. Then in 2016, Wang et al. constructed hyperbranched PAE (HPAE) via the facile A2 + B3 + C2 Michael addition strategy with amines and vinyl groups [9,10], which further improved the gene transfection performance of PAE vectors by increasing the optimal terminal group amount [9,11,12,13,14]. Despite years of research on HPAE structural factors, such as molecular weight, branch ratio, and terminal group screening [15,16,17,18], there is still a long pathway to clinical applications. To facilitate the development, we looked into the HPAE backbone structures and raised a question: have we dug out the full transfection potential of HPAE via the current synthesis strategy?

The branched structure construction was first intended to increase the pending terminal group amounts. Therefore, the terminal groups’ property in HPAE is still most noteworthy. Although hundreds of end-capped groups have been screened in previous work, the study of the effect of non-ideal terminal groups (the secondary amines from the monomers) on PAE is still blank (Figure 1). For the current HPAE synthesis strategy, the multifunctional monomers (branch monomers) were introduced to generate more terminal groups. The polymer molecular weight is controlled by the simple addition of excess-capped groups at specific time points [9,10,16,17,18,19,20,21]. However, the introduction of branched units results in large amounts of ideal end-capped and non-ideal terminal groups [22]. Considering the PAEs terminated with these groups exhibited poor transfection performance, [8] the existence of these terminal groups might hinder the gene delivery capability of HPAEs. Based on this understanding, we hypothesise that the transfection performance of HPAEs can be further improved by eliminating the non-ideal terminal groups. 

This paper used a new HPAE molecular weight control strategy to obtain PAEs with only ideal terminal groups. A series of HPAEs was achieved by adjusting the monomer vinyl/NH group ratios. In addition, the effect of non-ideal terminal groups was investigated with HPAEs of different molecular weights and different end-capped groups (primary amines and tertiary amines). 

## 2. Materials and Methods

### 2.1. Materials

For polymer synthesis and characterisation, commercially available amine monomer 1,4-butanediol diacrylate (BDA), 5-amino-1-pentanol (S5), and pentaerythritol tetraacrylate (PTTA) were purchased from Merck, Dublin, Ireland. 1-(3-aminopropyl)-4-methylpiperazine (E7) was purchased from Fisher Scientific, Dublin, Ireland. 1,5-Diamino-2-methyl pentane (DMP) was purchased from Fluorochem, Dublin, Ireland. Lithium bromide (LiBr) for GPC measurements was purchased from Merck, Ireland. Solvents dimethyl sulfoxide (DMSO), dimethylformamide (DMF), and diethyl ether were purchased from Fisher Scientific, Ireland. Deuterated chloroform (CDCl_3_) was purchased from Merck, Ireland. Sodium acetate (pH 5.2 ± 0.1, 3 M) was purchased from Merck, Ireland and diluted to 0.025 M before use. PicoGreen dsDNA Assay Kits were purchased from Thermo Fisher Scientific, Dublin, Ireland. The gWiz-GFP plasmids were purchased from Aldevron, USA. Cell culture media, trypsin-EDTA, penicillin/streptomycin, fetal bovine serum (FBS), and Hank’s balanced salt solution (HBSS) were purchased from Merck, Dublin, Ireland. The alamarBlue kit was purchased from Invitrogen, Dublin, Ireland. The commercial transfection reagent jetPEI was purchased from VWR, Dublin, Ireland. Lipofectamine 3000 (Lipo 3000) was purchased from BioSciences Ltd., Dublin, Ireland. jetOPTIMUS was purchased from VWR, Dublin, Ireland. All the agents were used according to the manufacturer’s protocols. HPAEs used in this study are proprietary to Branca Bunús Ltd., Dublin, Ireland.

### 2.2. Synthesis of HPAEs

HPAE A, B, C-E7, and A, B, C-DMP were obtained through different monomer feed ratios. The synthesis routes of B-E7 and B-DMP are described as an example below. BDA, PTTA, and S5 were dissolved in DMSO (30% *w*/*v*). The ratio of viny/NH was set as 1.2:1. The solution was bubbled with argon for 15 min, and the reaction occurred at 90 °C. When the molecular weight of the polymer no longer increased, the reaction was stopped by diluting the mixture to 10% *w*/*v* with DMSO. Then the reaction solution was equally separated into two parts. E7 or DMP was added to end-cap the acrylate terminated base polymers individually. After that, polymers were precipitated into diethyl ether and dried under a vacuum. For the A-E7 and A-DMP, the ratio of viny/NH was set as 1.15:1. For the C-E7 and C-DMP, the ratio of viny/NH was set as 1.25:1. 

HPAE C, D, E-E7, and C, D, E -DMP were obtained through conventional methods. BDA, PTTA, and S5 were dissolved in DMSO (30% *w*/*v*). The molar ratio of vinyl/NH was set as 1:1. The solution was bubbled with argon for 15 min, and the reaction occurred at 90 °C. When the molecular weight of the polymer reached the target value, the reaction was stopped by diluting the mixture to 10% *w*/*v* with DMSO. Then the reaction solution was equally separated into two parts. E or DMP was added to end-cap the acrylate terminated base polymers individually. After that, polymers were precipitated into diethyl ether and dried under a vacuum.

### 2.3. Molecular Weight Measurements

Number average molecular weight (*M*_n_), weight average molecular weight (*M*_w_), and polydispersity index (PDI) of polymers were determined by GPC equipped with a refractive index detector (RI). Specifically, the polymer sample was diluted to 5 mg/mL with DMF, filtered through a 0.2 µm filter and then measured by GPC. The columns were eluted with DMF and 0.1% LiBr at a flow rate of 1 mL/min at 60 °C. Columns were calibrated with linear poly(methyl methacrylate) (PMMA) standards.

### 2.4. Nuclear Magnetic Resonance (NMR) 

The polymer chemical structures and compositions were confirmed with one- and two-dimensional NMR spectra of ^1^H-NMR. Polymer samples were dissolved in CDCl_3_. Measurements were carried out on a Varian Inova 400 MHz spectrometer (Edinburgh, UK). Specifically, 100 µL of the reaction mixture was collected, diluted with 800 µL of deuterated solvent, and then measured by NMR. 

### 2.5. Polyplex Preparation

The polymer was dissolved in DMSO to 100 µL/µL stock solution and stored at −20 °C for the subsequent studies. For polyplex preparation, each polymer stock solution and DNA stock solution was dissolved in 25 mM sodium acetate (SA) to equal volume according to the polymer to DNA weight ratios (W/W). Then the polymer/SA solution was added to the DNA/SA solution, mixed well by pipetting, and incubated for 10 min at RT to formulate polymer/DNA complexes for the subsequent studies.

### 2.6. DNA binding Assay

PicoGreen dsDNA Assay Kits were used to quantify the DNA binding capacity of each polymer. Polyplexes were prepared as described above, added to a black 96-well plate with equal volume (100 µL) PicoGreen working solution and incubated for 5 min protected from light. Fluorescence was measured using a SpectraMax M3 plate reader (Molecular Devices, San Jose, CA, USA). DNA binding efficiency (BE) was calculated as Equation (1)
(1)BE=(FDNA−FSample)FDNA−FBlank 

F_DNA_ was the florescence measurement of free DNA without polymers, F_Sample_ was the florescence of a polyplex at a given weight ratio between polymer to DNA, and F_Blank_ was the florescence from PicoGreen working solution only with the buffer used for polyplex formulation.

### 2.7. Agarose Gel Electrophoresis

Polyplexes prepared as above were then loaded into 1% agarose gel wells, and electrophoresis was conducted at 120 V for 40 min.

### 2.8. Polyplex Size and Charge Characterisation and In Vitro Stability Assay

The size and zeta potential of polyplexes formulated by the polymer and gWiz-GFP plasmids were measured using Zetasizer Pro (Malvern Panalytical). The polyplex was prepared as described above and diluted to 1 mL of distilled water for measurement. The temperature of the samples was controlled at 25 °C. The in-vitro stability of polyplex was evaluated by their size changes after 4 h of incubation in the cell culture medium containing 10% fetal bovine serum. 

### 2.9. Cell Culture 

Human embryonic kidney 293 cells (HEK293) and rat adipose-derived stem cells (rADSC) were cultured in Dulbecco’s Modified Eagle Medium (DMEM) 6429 with 10% fetal bovine serum (FBS) and 1% penicillin/streptomycin and incubated at 37 °C and 5% CO_2_ in a humidified incubator. 

### 2.10. Cell Transfection

Cells were transfected with complexes prepared as described above, mixed with the cell culture medium, and added to cells at a DNA concentration of 5 µg/mL. The expression of the green fluorescent protein (GFP) was visualised 48 h after transfection using an Olympus IX81 fluorescence microscope (Olympus, Tokyo, Japan). The intensity of GFP fluorescence was analysed and semi-quantified using the ImageJ software (NIH, Bethesda, MD, USA). After imaging, the cell viability of treated cells was measured using the Alamar Blue kit according to the commercial manual. 

### 2.11. Statistical Analysis

All data are represented with the mean ± standard deviation (±SD), normally performing a minimum of three independent experiments. P values of less than 0.05 were considered significant (* *p* < 0.05, ** *p* < 0.01, *** *p* < 0.001). Statistical significance was reported in the figure legends. One-way ANOVA with posthoc Dunnett test was used to analyse the experiment’s results.

## 3. Result and Discussion

### 3.1. Synthesis and Characterisation of HPAEs

Previous research proved that the HPAEs could be easily synthesised via an “A2 + Bn + C2” Michael addition [9]. Specifically, diacrylates and diamines monomers reacted with multifunctional monomers in the solution. When HPAEs reached the target molecular weight, the reaction was stopped by adding excess-capped amines. To date, most of the HPAEs were constructed with a relatively high-branch monomer ratio, which means the reaction extents of vinyl and NH groups were still low when reaching the usual target molecular weights (*M*_w_~10–20 k Da). These low functional group-reaction extents indicated that although the unreacted vinyl groups were transformed into end-capped amines, many unreacted NH still existed, generating non-ideal termination (Figure 1). To decipher the effect of these non-ideal terminations, HPAEs with only ideal terminal groups (A, B, C-E7, and A, B, C-DMP, Figure 1) were achieved via increasing the ratio between vinyl/ NH groups. B4 (A2 type monomer), PTTA (B4 type monomer), and S5 (C2 type monomer) were used to construct the HPAE hyperbranched backbone. The feed ratios of vinyl/NH groups were set from 1.15 to 1.25. The reaction was allowed to carry on to eliminate the unreacted NH groups until the molecular weight no longer increased. Then specific end-capped groups, DMP (primary amines) or E7 (tertiary amines), were used to generate ideal terminations from excessive unreacted vinyl groups. In addition, HPAEs (D, E, F-E7 and D, E, F-DMP) with similar molecular weight but abundant non-ideal terminations (S5) were synthesised under a 1:1 ratio of vinyl and NH groups as comparisons.

GPC results displayed that A, B, C-E7 and D, E, F-E7; A, B, C-DMP and D, E, F-DMP had similar molecular weight and PDI. As shown in Table 1, A-E7, D-E7, A-DMP, and D-DMP all exhibited *M*_w_ around 7 k Da. The second molecular weight rank was B-E7, E-E7, B-DMP, and E-DMP, exhibiting *M*_w_ around 10 k Da. For the C-E7 and F-E7, the *M*_w_ was around 15 k Da. Furthermore, for the C-DMP, F-DMP, *M*_w_ was around 16~17 k. In addition, the molecular weight distributions of HPAEs with similar molecular weights had similar molecular weight distributions, as seen in Figure 2A,B. 

To analyse the ideal terminal group amount, the ideal terminal group ratios (TR) were analysed according to the NMR (Table 1). As expected, for the HPAE with similar molecular weight (Table 1 and Figure 2), those obtained from new methods exhibited more than twice the amounts of the ideal terminal groups compared to the conventional ones (A, B, C-E7 vs. D, E, F-E7). Figure 2C shows that the ideal terminal group signals in A, B, C-E7 are much stronger than the D, E, F-E7, respectively. A, B, C-DMP and D, E, F-DMP were end-capped from the identical HPAE backbones, thus expected to have similar ideal terminal group distributions as E7 end-capped HPAEs. It also proved that the conventional capped-stop molecular weight control method could result in HPAEs with many non-ideal terminations (S5), which could hinder their DNA package capacity.

Number average molecular weight (M_n_), weight average molecular weight (M_w_), and polydispersity Index (PDI) were GPC results. The ideal terminal group ratios (TR) were calculated as the molar ratio of E7/BDA according to the NMR.

### 3.2. Polyplex Characterisation of HPAEs

One of the essential requirements for efficient gene delivery is the effective condensation of DNA to form nano-sized and positively charged polyplexes. Herein, E7 end-capped HPAEs were assessed to explore the effect of non-ideal terminations in the DNA package. The interactions of HPAEs with DNA were analysed by dynamic light scattering (DLS) and Picogreen assay. As can be seen in Figure 3A, although all HPAEs exhibited higher than 80% DNA binding efficiencies, the A, B, and C-E7 displayed slightly more robust DNA binding capabilities than their corresponding HPAEs D, E, and F-E7. The high DNA binding capabilities of HPAEs were further confirmed by agarose gel electrophoresis (Appendix A). According to the DLS assessment (Figure 3B), all HPAEs had sufficient capabilities to form nano-sized polyplexes. Notably, for those polyplexes based on A, B-E7, the polyplex sizes were larger than those of D, E-E7, which could be attributed to the increasing number of E7 groups. The growing size of the polyplex will benefit micropinocytosis behaviour, possibly enhancing cellular uptake [23]. In contrast, for the HPAEs with higher molecular weight, the size of the polyplex formed by C-E7 was slightly smaller than the F-E7-based polyplex, which is because, for the HPAEs with high molecular weight, the polyplex formation is dominated by HPAEs’ hydrophobic properties instead of the end-capped amount. Therefore, F-E7 with more hydrophilic S5 groups helps reduce the polyplex size. Further, A, B, and C-E7-based polyplexes have been selected and demonstrated to have better serum stability than D, E, and F-E7-formulated polyplexes, respectively. As seen in Appendix A, after 4 h of incubation in serum, the sizes of polyplex formulated by D, E, and F-E7 changed more significantly compared to their counterparts. To enable a robust cellular uptake, the cationic polymer-based polyplex should maintain a positive surface charge to enhance the interaction between cell membranes. The zeta potential assessment in Figure 3C indicated that the non-ideal terminations impeded the formation of the polyplex positive charge surface due to the lack of protonated nitrogen groups. In all comparison groups with similar molecular weight, the polyplexes with only ideal terminations (A, B, C-E7) exhibited higher zeta potential than those of D, E, F-E7 based polyplexes. Considering that the cellular uptake process will last for hours, the polymer vector must have the capabilities to package DNA into the polyplex, maintain the polyplex structure, and protect the DNA before cellular uptake. Therefore, the DNA binding ability of HPAEs was assessed after 4 h of incubation at 37 °C (Figure 3D). For all HPAE groups, their polyplexes all maintained high DNA binding efficiencies (≥80%), indicating that the existence of non-ideal terminations (S5) will not affect their DNA protection capabilities.

### 3.3. In Vitro Transfection of HPAEs

Non-viral gene delivery vectors aim to maximise transfection efficiency while minimising cytotoxicity. Gene transfection efficiency and cytotoxicity of HPAEs were evaluated with a reporter DNA plasmid expressing GFP in HEK293 cells. For all the tested polymer/DNA ratios, HPAEs with only ideal terminations (A, B, C-E7 and A, B, C-DMP) exhibited much better gene transfection performance (more than twice fluorescence intensity) compared to corresponding controls (D, E, F-E7 and D, E, F-DMP) (Figure 4A, B). The same tendency was confirmed in rADSC, while the transfection efficacies were significantly higher with B- and C-E7, DMP (Appendix A). The A-E7 and A-DMP did not present robust efficacy, which also demonstrated that the molecular weight of HPAEs plays a key role, especially when the cells are difficult to transfect. Meanwhile, for both E7 and DMP end-capped HPAEs, transfection performance increased as their molecular weight increased, which agrees with previous studies. HPAEs with only ideal terminations offered more end-capped groups, leading to larger polyplex size and higher zeta potential, indicating better cellular uptake, more effective endosome escape, and subsequently higher transgene expression. Regarding cytotoxicity (Figure 4C,D and Appendix A), all HPAEs preserved high (over 75%) cell viability even at the highest *w*/*w* ratio of 30:1. These results further illustrate that, by eliminating the non-ideal terminations of HPAEs, we can significantly enhance transfection performance while maintaining high biocompatibilities.

Well-studied commercial transfection reagents Lipo 3000, jetPEI, and jetOPTIMUS were then used as positive controls to evaluate the transfection performance of the optimal HPAEs (C-E7 and C-DMP). As can be seen in Figure 5, HPAEs displayed more robust GFP expression than the commercial transfection agent. These results indicated that the gene delivery capability of HPAEs can be significantly enhanced by eliminating the non-ideal terminations. On the other hand, there have been many other types of gene delivery materials developed. Especially lipid-based systems have been extensively developed and applied in the real world for clinical uses. Numerous lipid structures with potent in vitro and in vivo transfection efficacy have been generated via combinatorial reaction schemes, including C12–200, 503O_13_, 306O_i10_, OF-02, TT3, 5A2-SC8, SM-102 (Moderna vaccine), and ALC-0315 (Pfizer–BioNTech vaccine) [24]. Besides lipid-based vectors, the specific biological and chemical functions of other candidates, including peptide/protein-based materials (i.e., PepFect14 [24], protamine [25], and virus-like protein PEG10 [26]) and cationic nanoemulsions (i.e., squalene [24]), for genetic material delivery have been further explored. For those systems, the structures of cargo binding groups and numbers and formulation ratios of those critical structures have been found to be vital for gene delivery efficacy, thus sharing the same principles with the key finding of this study. 

## 4. Conclusions

This work studied the effect of non-ideal terminations in HPAEs for the first time. The existence of secondary amine residues on the backbone monomer proved to hamper the DNA package and hinder efficient transfection. To eliminate the non-ideal terminations and further explore the hyperbranched structure merits of HPAE, a functional group ratio adjustment strategy was used to control HPAE molecular weight. In addition, several highly efficient HPAEs with new structures were synthesised. By eliminating the non-ideal terminations, their transfection performances were better than the three commercial benchmark transfection reagents (Lipo3000, jetPEI, and jetOPTIMUS). This work fulfilled an essential research blank in HPAE vector development, providing insights into non-viral gene delivery nanostructured material development.

## Data Availability

Not applicable.

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
