# Peer review of "Hyperbranched Poly(β-amino ester)s (HPAEs) Structure Optimisation for Enhanced Gene Delivery: Non-Ideal Termination Elimination"

_nanomaterials, 2022, doi:10.3390/nano12213892_

Round 1

Reviewer 1 Report

The paper by Li et al. deals with the optimization of poly(β-amino ester)s for gene delivery. The topic of worthy of investigation and well fits with the aim and scope of nanomaterials. Overall, the manuscript is well organized and should be eventually published. Nevertheless, this reviewer is suggesting to address some issues before further processing of the paper.

Below some comments as per article sections.

Abstract. in its present form it seems too literal and closely to an introduction section. Authors should consider the possibility to revise the section by inserting some key result data here.

Introduction. Authors well presented the aim of their work and the previous knowledge in the specific field. This reviewer has no specific comments to this section.

Materials and Methods. The experimental set-up is well designed and performed. This reviewer has no specific comments to this section. Few comments are related to the need to insert city and country of the company where reactants were purchased (as per MDPI rules), as well as to number the equations.

Results and Discussion. Results were well presented, but a comparison with different systems available in the literature can help to strengthen the key findings of the study. Authors should consider this as an improvement of the discussion of their data

Conclusion. Authors should consider the real need of the last sentence of the section (lines 306-308)

Author Response

Dear Reviewer,

Thank you so much for your generous suggestions and comments. We have read and carefully revised our manuscript according to your suggestions. The changes we made are tracked in the text for your easy reference (Please see the attachment).

The followings are our point-by-point responses to your comments and suggestions.

(1) Abstract. in its present form it seems too literal and closely to an introduction section. Authors should consider the possibility to revise the section by inserting some key result data here.

Response: Many thanks to the Reviewer for the suggestion. We have revised the manuscript's abstract and inserted some key result data (lines 31-34).

(2) Materials and Methods. The experimental set-up is well designed and performed. This Reviewer has no specific comments to this section. Few comments are related to the need to insert city and country of the company where reactants were purchased (as per MDPI rules), as well as to number the equations.

Response: We thank the Reviewer's good suggestion. We have added the locations of purchases in the materials and methods section (lines 98-114).

(3) Results and Discussion. Results were well presented, but a comparison with different systems available in the literature can help to strengthen the key findings of the study. Authors should consider this as an improvement of the discussion of their data.

Response: Indeed, thanks to the Reviewer for pointing it out. The current HPAE polymers are designed according to standard poly(beta-amino ester)s (PAEs), which have been developed broadly, and we have included the introduction and comparison or combination of PAEs with other systems, including lipid-based, protein-based and other types in our discussion section (line 308-319).

(4) Conclusion. Authors should consider the real need of the last sentence of the section (lines 306-308).

Response: Many thanks for the Reviewer's suggestion. We have removed the last sentence of the Conclusion section and left the choices of future applications open for readers (line 335).

Reviewer 2 Report

Comments on nanomaterials-1987911

In this manuscript, the authors developed hyper-branched poly(β-amino ester)s (HPAEs) as safe and efficient gene delivery vectors. A series of characterizations and in vitro studies were performed to study the chemical structures, nanoparticle morphology, gene transfection efficiency, and cell viability. Overall, this is good work, and the conclusions were demonstrated by the data. However, the authors need to address several issues.

1.       The statistical analysis method needs to be described clearly (e.g., Student’s t-test or one-way ANOVA).

2.       Gel electrophoresis study should be performed to study the complex of HPAEs with plasmid DNA at different ratios.

3.       Flow cytometry should be performed to quantify GFP+ cells and GFP intensities after transfection. Fluorescent imaging cannot provide quantitative analysis.

4.       The transfection was only tested in HEK293 cells. It should also be tested in at least one type of hard-to-transfect cell line.

5.       The in vitro stability of the HPAE-DNA complex in serum should be studied. 

Author Response

Dear Reviewer,

Thank you so much for your generous suggestions and comments. We have read and carefully revised our manuscript according to your suggestions. The changes we made are tracked in the text for your easy reference (Please see the attachment).

The followings are our point-by-point responses to your comments and suggestions.

(1) The statistical analysis method needs to be described clearly (e.g., Student's t-test or one-way ANOVA).

Response: Thanks a lot to the Reviewer for pointing it out. We have revised the manuscript and clarified the statistical analysis method (lines 189-190).

(2) Gel electrophoresis study should be performed to study the complex of HPAEs with plasmid DNA at different ratios.

Response: Many thanks to the Reviewer. We have done the additional gel electrophoresis study and included it in the Result and Discussion section (lines 247-248) and supporting information (Figure S1).

(3) Flow cytometry should be performed to quantify GFP+ cells and GFP intensities after transfection. Fluorescent imaging cannot provide quantitative analysis.

Response: Thanks a million to the Reviewer for this suggestion. The GFP expression results performed in Figure 4 and Figure 5 were semi-quantitative analyses using ImageJ, a convenient standard method for efficacy comparison. The bar graph shows that the differences between groups were significant and sufficient to support our study.

Indeed, it is an excellent suggestion to further quantitatively study the transfection efficacy. Unfortunately, we had limited access to flow cytometry before the deadline for resubmission, and we will perform flow cytometry in the follow-up study as suggested. We hope the Reviewer finds our responses satisfactory. Thanks again for this valuable suggestion.

(4) The transfection was only tested in HEK293 cells. It should also be tested in at least one type of hard-to-transfect cell line.

Response: Many appreciations to this comment. We have selected another cell line, rADSC, a type of stem cells that is hard to transfect, and representative HPAE polymers to perform the additional test and have described it in the Result and Discussion section (lines 284-287) and supporting information (Figure S3 and S4).

(5) The in vitro stability of the HPAE-DNA complex in serum should be studied.

Response: We thank the Reviewer's good suggestion. We selected to study the size change of polyplex nanoparticles in serum to perform the additional in vitro stability tests, and we have included it in the Result and Discussion section (lines 257-260) and supporting information (Figure S2).

Round 2

Reviewer 1 Report

Authors addressed all concerns and revised the manuscript accordingly. This reviwer is recommending publication of the manuscript in its current form

Reviewer 2 Report

The authors have addressed all my comments in the revised manuscript. Therefore, I recommend acceptance of this manuscript.